# MRI risk factors for development of avascular necrosis after closed reduction of developmental dysplasia of the hip: Predictive value of contrast-enhanced MRI

**Jung-Eun Cheon** [1,2,3], **Ji Young Kim**[1,2,3¤], **Young Hun Choi**[1,2,3], **Woo Sun Kim**[1,2,3], **Tae-Joon Cho**[4,5], **Won Joon Yoo** [4,5]*

1 Department of Radiology, Seoul National University College of Medicine, Seoul, Korea, 2 Department of Radiology and Institute of Radiation Medicine, SNUMRC (Seoul National University Medical Research Center), Seoul National University Hospital, Seoul, Korea, 3 Division of Pediatric Radiology, Seoul National University Children's Hospital, Seoul, Korea, 4 Department of Orthopedic Surgery, Seoul National University College of Medicine, Seoul, Korea, 5 Division of Pediatric Orthopedics, Seoul National University Children's Hospital, Seoul, Korea

¤ Current address: Department of Radiology, Seoul National University Bundang Hospital, Seongnam, Korea
* yoowj@snu.ac.kr

**Data Availability Statement:** All relevant data are within the paper.

## Abstract

### Purpose

The purpose of this study was to identify imaging risk factors on contrast-enhanced hip MRI after closed reduction of developmental dysplasia of the hip (DDH) that could predict future development of avascular necrosis (AVN) of the femoral head.

### Materials and methods

Fifty-eight infants (F: M = 53: 5, aged 3–18 months) who underwent immediate postoperative contrast-enhanced hip MRI after closed reduction of DDH were included in this study. Quality of reduction (concentric vs eccentric reduction with or without obstacles), abduction angle of the hip, presence of ossific nucleus, and pattern of contrast enhancement of the femoral head were retrospectively evaluated on MRI. Interobserver agreement of contrast enhancement pattern on MRI were evaluated by two radiologists. Development of AVN was determined through radiographic findings at 1 year after reduction.

### Results

AVN of the femoral head developed in 13 (22%) of 58 patients. Excessive abduction of the hip joint (OR 4.65, [95% CI 1.20, 18.06] and global decreased enhancement of the femoral head (OR 71.66, [95% CI 10.54, 487.31]) exhibited statistically significant differences between the AVN and non-AVN groups (P < 0.05). Eccentric reduction (P = 0.320) did not show statistically significant difference between two groups and invisible ossific nucleus (P = 0.05) showed borderline significance. Multi-variable logistic regression indicated that global decreased enhancement of the femoral head was a significant risk factor of AVN (OR

**Funding:** The authors received no specific funding for this work.

**Competing interests:** The authors have declared that no competing interests exist.

27.92, 95% CI [4.17, 350.18]) (P = 0.0031). Interobserver agreement of contrast enhancement pattern analysis and diagnosis of AVN were good (0.66, 95% CI [0.52, 0.80]).

## Conclusion

Contrast-enhanced hip MRI provides accurate anatomical assessment of the hip after closed reduction of DDH. Global decreased enhancement of the femoral head could be used as a good predictor for future development of AVN after closed reduction of DDH.

## Introduction

Closed reduction and spica casting under general anesthesia is a common treatment for infants with developmental dysplasia of the hip (DDH) who fail initial nonsurgical management or those who present late [1–3]. The aim of closed reduction is to obtain a stable and concentric reduction of the hip while limiting the risk of one of the most devastating complications, avascular necrosis (AVN) of the femoral head [4–7]. AVN may lead to femoral head deformity, persistent acetabular dysplasia, abnormal growth of the femoral neck and greater trochanter, coxa vara, limb length discrepancy, and eventual osteoarthritis in adult life [3,4,8].

The reported incidence of AVN has varied widely from 6% to 47% [4,7,9]. This may be attributed to the lack of consensus on the specific radiographic features that define AVN in DDH. The most widely-used classification of AVN is that from Salter et al. [10] who defined five characteristic radiographic findings at 1 year after reduction. Several factors such as excessive hip abduction, intra-articular obstacles that block concentric reduction, absence of an ossific nucleus, and age at reduction have been postulated to associate with development of AVN [7,8,11,12].

Magnetic resonance imaging (MRI) has become the preferred technique in clinical settings after hip reduction and casting to confirm the concentric reduction critical for normal development of the femoral head and remodeling of the acetabulum [13–16]. Recently, several studies have reported the use of contrast-enhanced MRI to detect abnormal enhancement of the femoral head as surrogate markers for predicting AVN [9,17,18]. Early detection of ischemia at the time of hip reduction and possible prediction of later development of AVN would be a basis for further studies on modification of current treatment or creation of new treatment methods. The purpose of this study is to review contrast-enhanced hip MRI after closed reduction of DDH and to identify imaging risk factors that could predict development of avascular necrosis (AVN) of the femoral head.

## Materials and methods

### Patients

This study was approved by the Institutional Review Board of Seoul National University Hospital (IRB No. H-2003-060-1108), and the requirement for informed consent was waived. A list of 77 patients who underwent contrast-enhanced MRI under the predetermined protocol after closed reduction of DDH at a single tertiary-care pediatric center from March 2005 to December 2014 was collected. Patients were included in this study according to the following eligibility criteria: (a) Patients who were diagnosed with unilateral or bilateral DDH based on ultrasonography and plain radiography, (b) patients aged 18 months or younger who were treated by closed reduction at the Seoul National University Children's Hospital, (c) contrast-

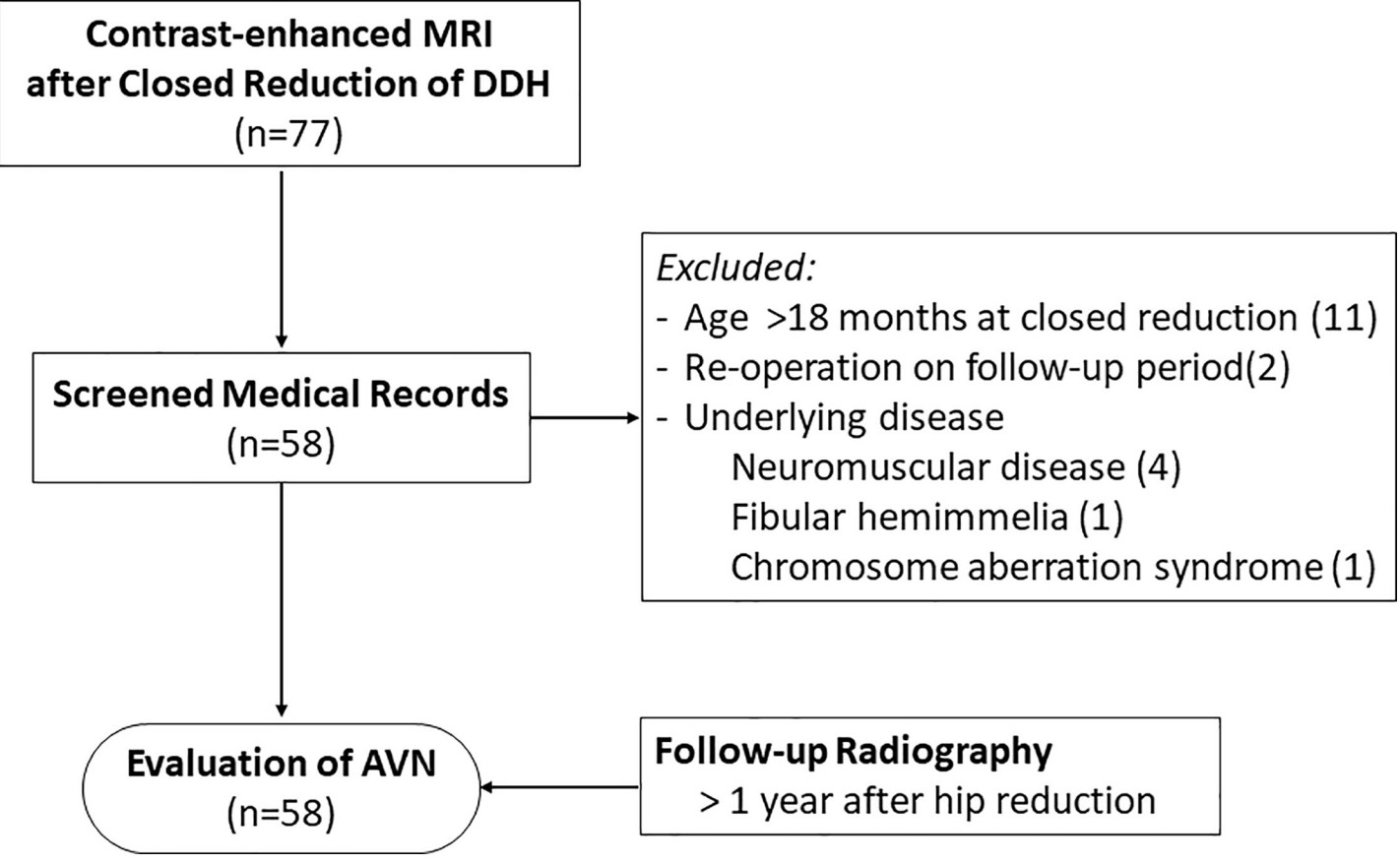

**Fig 1. Flow chart of the study inclusion.**

enhanced hip MRI performed within 24 hours of closed reduction and spica cast application, and (d) follow-up hip plain radiographs taken in 1 year after closed reduction. Eleven patients who were older than 18 months at the time of hip MRI, 4 patients with neuromuscular disease, 1 patient with ipsilateral fibular hemimelia, and 1 patient with chromosomal aberration syndrome were excluded from the study. Two patients with initial closed reduction for DDH and conversion to open reduction during the follow-up period were also excluded (Fig 1). A total of 58 patients was included in this study. Fifty-three of these patients (91%) were female and five (9%) were male. Mean age at closed reduction was 10.95 ± 4.65 months (range: 3–18 months). Unilateral hip was involved in 54 patients (left– 36 patients, right– 18 patients), while bilateral hips were involved in four patients. Ten patients had undergone Pavlik harness or hip abduction orthosis treatment prior to the closed reduction procedure. Preoperative skin traction was performed in 33 patients. Twenty-nine patients underwent percutaneous adductor tenotomy at the time of closed reduction. The mean postoperative follow-up duration was 68.50 ± 28.09 months with minimum follow-up of 26 months (range: 26–134 months).

## MR imaging protocol

MR imaging was performed with a 1.5-MR unit at our institution (Signa Excite; GE Healthcare; Milwaukee, WI, USA or Magnetom Avanto; Siemens, Erlangen, Germany). MR imaging protocol comprised the following sequences: axial T2-weighted image with fat saturation, T1-weighted image, gradient echo image (GRE) or multiple-echo data image combination

(MEDIC) sequence, and axial/coronal post-contrast T1-weighted image with fat saturation. Post-contrast T1-weigted imaging was performed after injection of a standard dose of 0.1 mmol/kg gadoterate meglumine (Dotarem; Guerbet) at a rate of 1–1.5 mL/sec using an MR imaging–compatible power injector (Spectris Solaris; Medrad). A bolus of contrast material was followed by a 5-mL bolus of saline that was administered at the same injection rate. Post-contrast T1-weighted images were obtained in 2 minutes after injection of contrast materials. Total imaging times were approximately 25 minutes. All infants were sedated using chloral hydrate (30–50 mg/kg body weight) for the MRI examination. Pulse oximetry was performed to continuously monitor arterial hemoglobin oxygen saturation (SpO2) and heart rate. Airway patency was monitored intermittently by specially trained pediatric nurses.

## MR imaging analysis

All hip MRIs were retrospectively interpreted by a pediatric radiologist (C.J.E., 22-years of experience). Image interpretation was aimed at (a) confirming quality of reduction (concentric reduction vs eccentric reduction) with identifying obstacles that limit concentric reduction, (b) assessing hip abduction angle, and (c) evaluating contrast enhancement of the proximal femoral epiphysis after reduction as well as presence of ossific nucleus at the time of reduction. For interrater agreement of contrast enhancement pattern analysis, the 2nd reader (C.Y.H., 11-years of experience) independently reviewed contrast-enhanced MR images.

**Quality of reduction.** Concentric reduction was assessed at the level of the triradiate cartilage. On MRI, the cartilaginous epiphysis or ossific nucleus of the femoral head should lie anterior to the line joining the triradiate cartilages. The medial joint space width on axial images is defined as the distance between the triradiate cartilage and femoral capital epiphysis and was measured (Fig 2). If the medial joint space width was greater than 3 mm, it was considered eccentric reduction [19,20].

Obstacles that limit concentric reduction of the femoral head, such as the pulvinar and inversion of the hypertrophic acetabular labrum, were evaluated. The pulvinar is a layer of fibrofatty tissue filling the non-articular floor of the acetabulum (Fig 3A). A normal acetabular labrum is roughly triangular in shape, with a down-tilted, lateral orientation on coronal images, demonstrating low signal intensity on both T1- and T2-weighted images. In DDH patients, the labrum is hypertrophied and often inverted into the acetabulum, which causes entrapment between the femoral head and acetabulum and disturb deep reduction of the femoral head. The neolimbus is a ridge of proliferative fibrocartilage within the superolateral aspect of the acetabulum, which is often adjacent to the hypertrophied labrum (Fig 3B).

**Abduction angle of the hip.** Each reader measured the abduction angle of the hip on axial GRE or MEDIC image at the level of the triradiate cartilage. Hip abduction angle is defined between the line along the mid-femoral shaft and the line connecting the posterior ischial tuberosity. An abduction angle greater than 60° was considered inappropriate excessive abduction [21] (Fig 4).

**Proximal femoral epiphysis.** Presence of ossific nucleus was evaluated on axial GRE or MEDIC images. Contrast enhancement of the femoral epiphysis was visually graded at both the axial plane at the level of the triradiate cartilage and at the mid-coronal plane as normal, a geographic enhancement, or a global decreased enhancement, i.e., normal (striated or speckled pattern of enhancement in the cartilaginous femoral epiphysis with no contrast enhancement defect area), geographic enhancement (heterogeneous enhancement with area of decreased enhancement less than < 75% of the proximal femoral epiphysis (3/4 of femoral head), and global decreased enhancement (diffusely decreased enhancement of the femoral head ≥ 75% of the femoral epiphysis with near complete loss of the striated or speckled pattern of

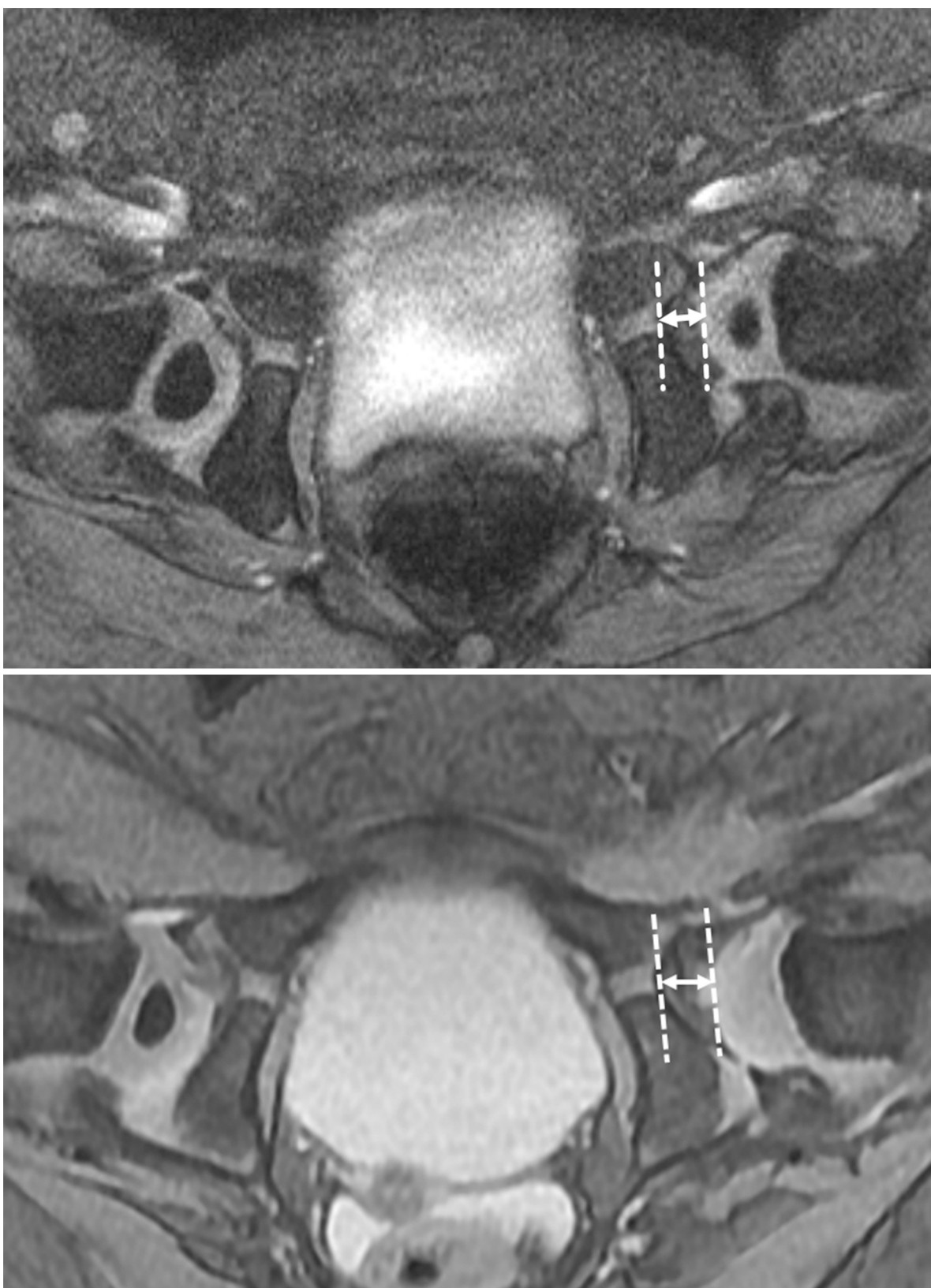

**Fig 2. Femoral head position: Concentric reduction vs. eccentric reduction.** A. Medial joint space distance is defined as the distance between the triradiate cartilage and femoral capital epiphysis. If the medial joint space width is 3 mm or less, it is defined as a concentric reduction. B. Eccentric reduction of the left DDH in a 17-month-old girl. Medial joint space distance is 6 mm. Note the dysplastic left acetabulum and no visible ossification center of the left proximal femoral epiphysis.

enhancement in the cartilaginous femoral epiphysis) (Fig 5). If there were disagreement between two readers, final interpretation of contrast enhancement pattern of the proximal femoral epiphysis was obtained in consensus.

## Reference standard for AVN

Post-reduction radiographs obtained at or around 1 year after closed reduction and the latest post-reduction radiograph after 1 year were retrospectively reviewed to identify the development of AVN. One pediatric radiologist (K.J.Y., with 8-years of experience) and one pediatric orthopedic surgeon (Y.W.J., with 22-years of experience) independently reviewed post-reduction radiographs. AVN was diagnosed using the Salter's criteria [10]: (1) Failure of appearance of the ossific nucleus within 1 year after reduction, (2) failure of growth in the ossific nucleus within 1 year after reduction, (3) proximal femoral metaphyseal widening, (4) epiphyseal fragmentation, and (5) residual deformity of the femoral head/neck. The reviewers were blinded to the hip MR imaging results. If the diagnosis of AVN was discordant, final diagnosis was determined using a combination of Salter criteria and Kalamchi and MacEwen classification in consensus.

## Statistical analysis

Potential candidate risk factors for AVN considered in this study were clinical features including demographical characteristics and MRI features. The associations between imaging characteristics and AVN status were evaluated using penalized logistic regression to reduce sparse data bias [22]. In patients with bilateral DDH, right side hip data were selected to avoid separation issue with clustered data. MR imaging features related to reduction status of DDH (concentric or eccentric reduction with or without excessive hip abduction) and features associated with proximal femoral epiphysis (presence of ossific nucleus and contrast enhancement of proximal femoral epiphysis). Contrast enhancement patterns were categorized two groups: normal or geographic enhancement vs global decreased enhancement. Categorical variables are reported as proportion (numerator/denominator and percentage) with associated effect size (odds ratio [OR]) and 95% confidence interval (CI). All continuous variables are presented with a measurement of central tendency (mean or median) and spread (standard deviation (SD) or range). A P-value less than 0.05 was considered to be statistically significant, and a 95% CI was reported for each variable. The multi-variable model was constructed based on the clinical importance and findings from uni-variable analysis. Intra-articular obstacles preventing complete reduction of DDH such as pulvinar were not evaluated in multi-variable analysis because controversy exists regarding potential of resolving soft tissue interposition with time [12] or small number of cases such as labral inversion. The optimism corrected area-under-the-curve was calculated using 1000 bootstrap samples. All statistical analyses were performed with statistical software (SAS 9.4, SAS Institute, Inc.; Cary, NC).

Interrater agreement of imaging diagnosis of AVN was analyzed with the Cohen κ statistic, and interrater agreement of contrast enhancement pattern was analyzed with weighted kappa statistics. The κ statistic was interpreted as follows: less than 0–0.20, slight agreement; 0.21–0.40, fair agreement; 0.41–0.60, moderate agreement; 0.61–0.80, substantial agreement; and 0.81–1.0, almost perfect agreement.

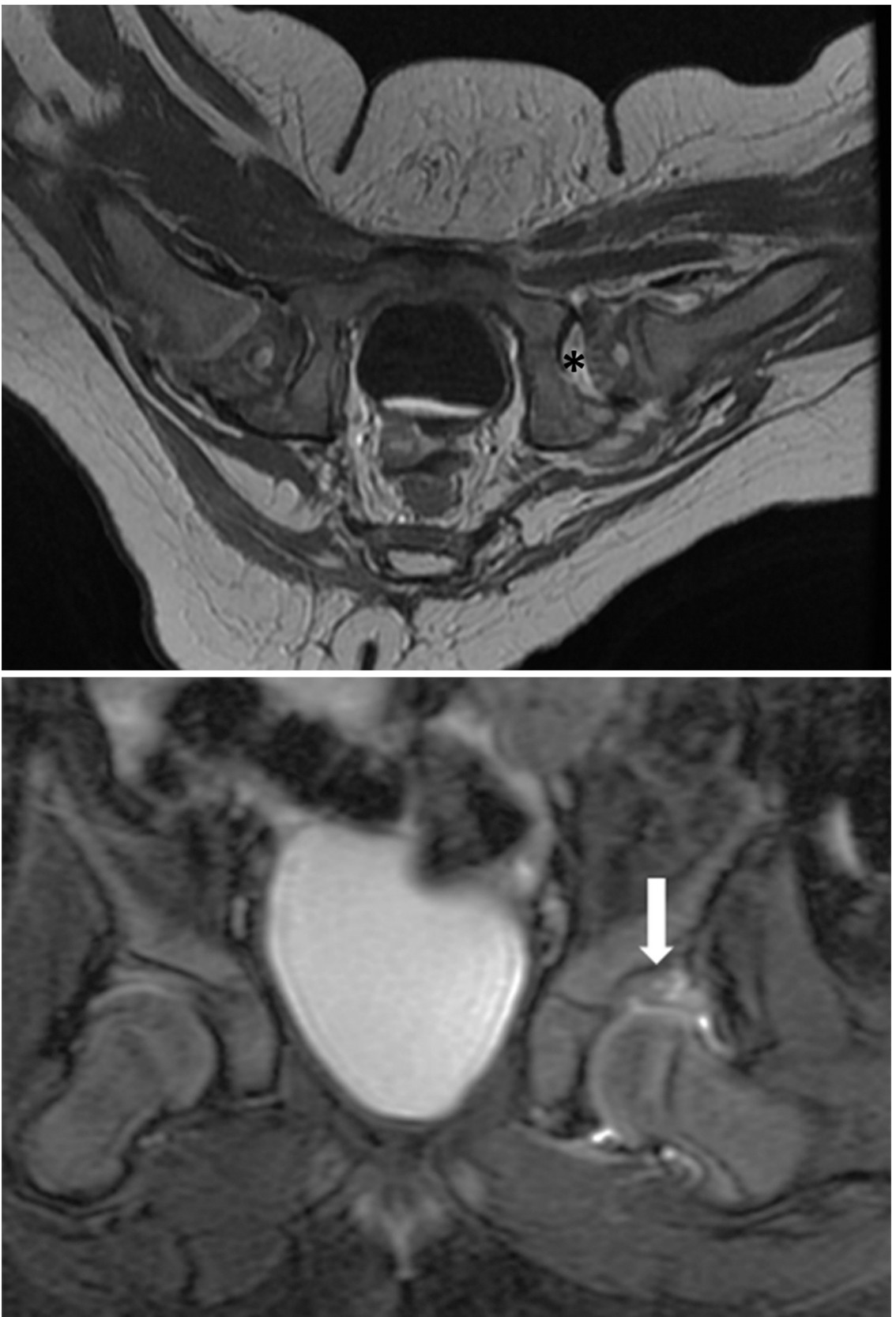

**Fig 3. Obstacles of closed reduction: MR features.** A. The contrast-enhanced axial T1-weighted image shows the pulvinar (asterix) as a thick fibrofatty tissue blocking concentric reduction in the left hip joint space. Note the punctate enhancement of the left proximal femoral epiphysis. B. Coronal T2-weighted with fat-suppression obtained after closed reduction of the left hip shows triangular shaped, hyperintense soft tissue at the chondrolabral junction of the left acetabulum (arrow) reflecting the neolimbus.

## Results

### Clinical characteristics

AVN of the femoral head developed in 13 of 58 patients (22%). The left hip was involved in 7 patients and the right hip in 6 patients. Interrater agreement was almost perfect between the two readers (Kappa 0.90, 95% CI [0.77, 1.00]). There were no significant differences in the distribution of age (p = 0.982) or sex (p = 0.337) between the AVN and non-AVN groups. There was no significant difference in mean follow-up period between the AVN and non-AVN groups (p = 0.749). There were no significant differences in development of AVN in patients with breech presentation (p = 0.680), oligohydramnios (p = 0.320), or family history of DDH (p = 0.898). No significant differences in development of AVN were seen in patients who had undergone prior Pavlik harness or hip abduction orthosis treatment (p = 0.841) or based on performance of adductor tenotomy (p = 0.124) during closed reduction (Table 1).

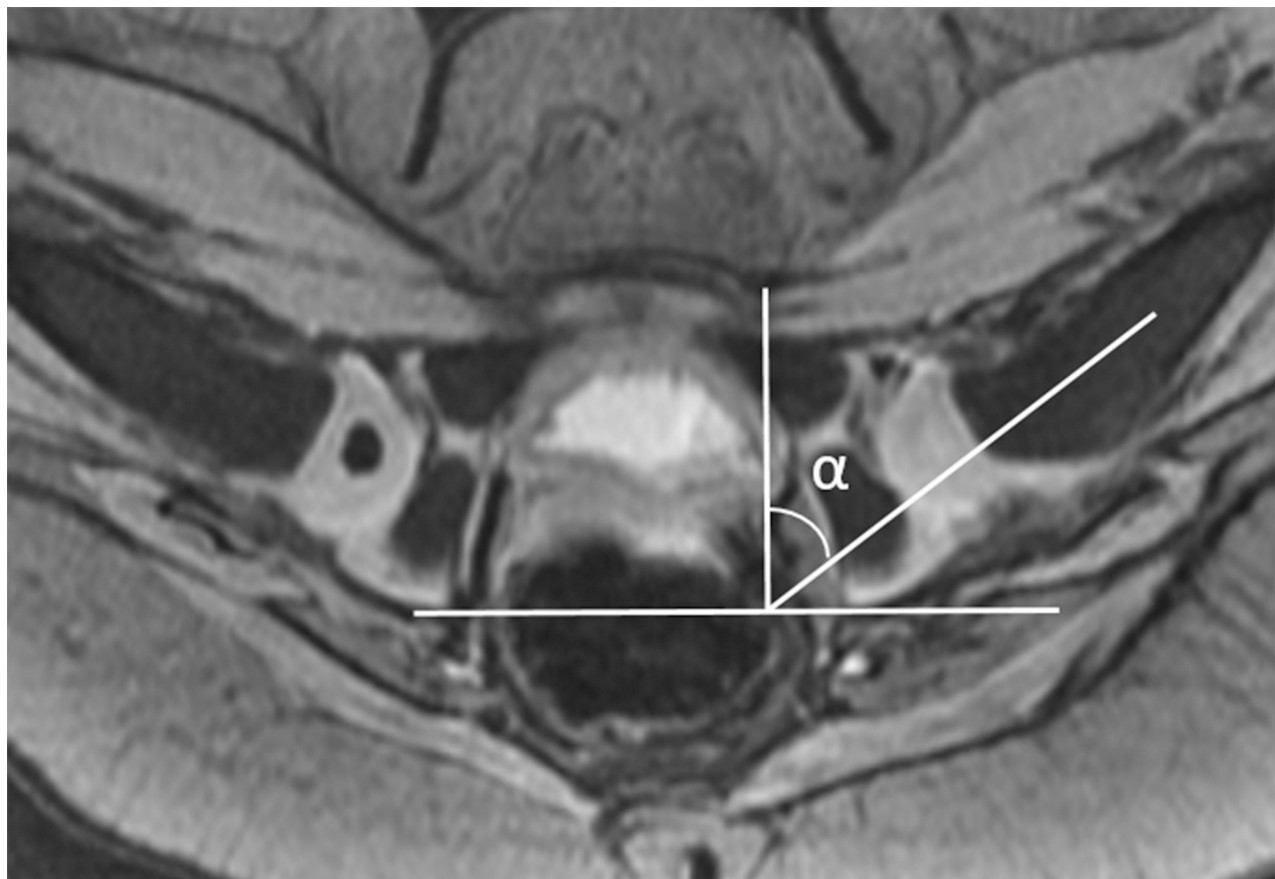

**Fig 4. Abduction angle measurement on MRI.** Abduction angle is measured on the MEDIC image. Hip abduction angle is defined as the angle between the line along the mid-femoral shaft and the line connecting the posterior end of the ischial tuberosities (α angle).

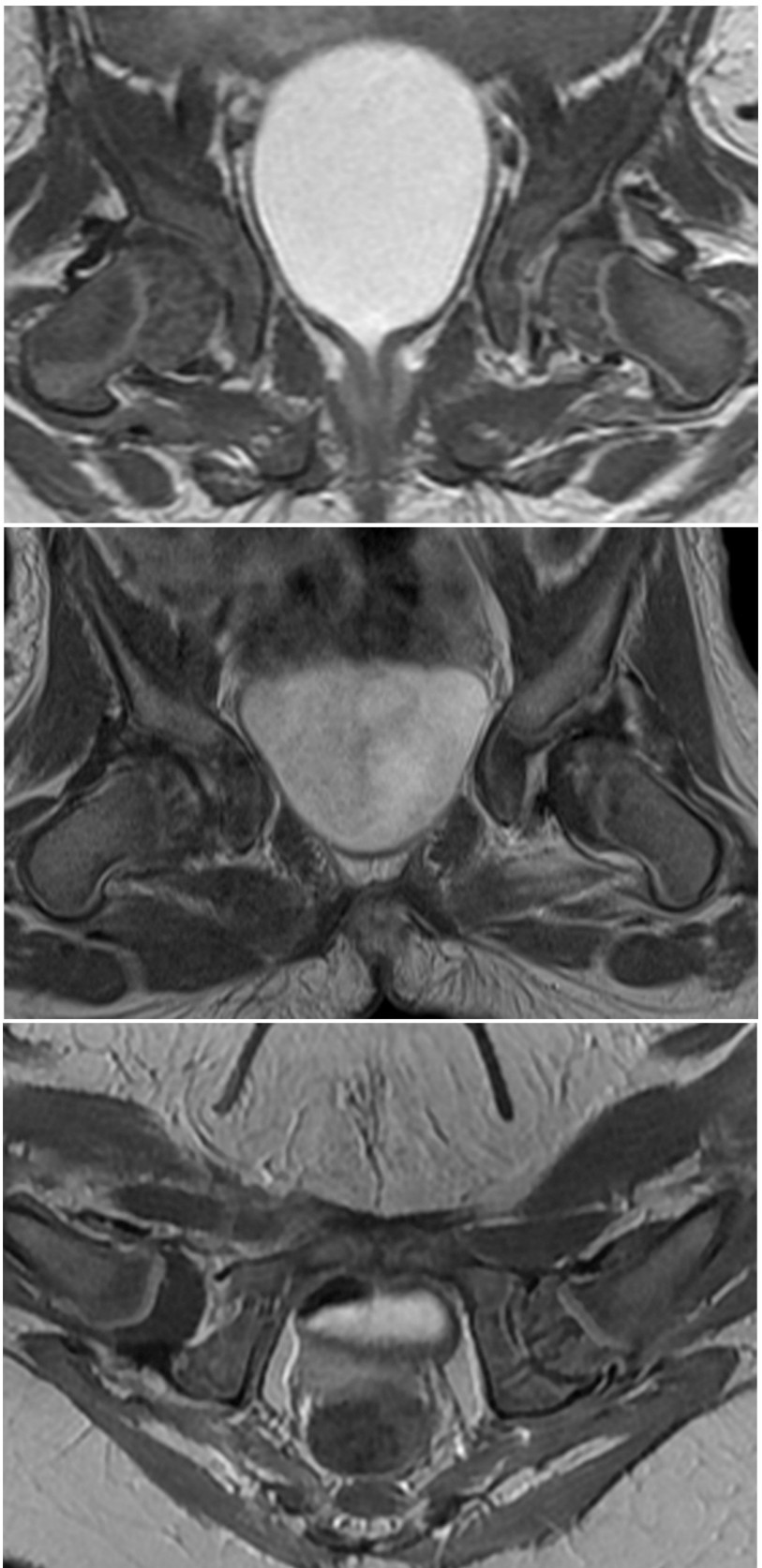

**Fig 5. Pattern of contrast enhancement of the proximal femoral epiphysis.** A. Contrast-enhanced coronal T1-weighted image in a 5-month old girl after closed reduction of the left hip shows a normal striated pattern of enhancement in the cartilaginous femoral epiphysis. B. Contrast-enhanced coronal T1-weighted image in a 4-month-old girl after closed reduction of the left hip shows a geographic area of decreased enhancement in the left femoral head. C. Contrast-enhanced axial T1-weighted image in a 7-month-old girl after closed reduction of the right hip shows a global decreased enhancement of the right femoral head.

## MRI parameters

**Quality of reduction.** Mean value of medial joint space distance was $4.64 \pm 1.32$ mm (range: 2.0–8.0 mm). Deep concentric reduction was observed in 10 of 58 hips (17%), and the other 48 hips showed eccentric reduction (> 3 mm in medial joint space distance). Medial joint space distance was wider in the AVN group ($5.46 \pm 1.27$ mm) compared to the non-AVN group ($4.47 \pm 1.26$ mm) ($p = 0.014$) (Fig 2). In 13 patients of AVN, 12 patients (92%) exhibited eccentric reduction of the hip whereas 45 patients of non-AVN showed eccentric reduction of the hip in 36 patients (80%) ($p = 0.320$). Logistic regression analysis demonstrated that eccentric reduction exhibited an increased odds ratio of 3.00 [95% CI: 0.34, 26.18].

Obstacles that limit concentric reduction were detected as follows: pulvinar in 50 patients (86%), neolimbus in 37 patients (64%) and labral inversion in two patients (3%). All patients of AVN showed pulvinar whereas 82% (37/45) of non-AVN showed pulvinar on MRI ($p = 0.251$) (Fig 3). Logistic regression analysis demonstrated that presence of pulvinar exhibited an increased odds ratio of 6.12 [95% CI: 0.28, 134.27]. Neolimbus was detected in 37 patients (64%): 69% (9/13) of AVN vs 62% (28/45) of non-AVN ($p = 0.644$). Labral inversion was not detected in patients with AVN whereas two patients with non-AVN showed labral inversion ($p = 0.819$).

**Abduction angle of the hip.** Mean hip abduction angle was $56.32 \pm 6.41$ (range: 38–70). Excessive hip abduction (> 60˚) was found 13 patients (22%): 46% (6/13) of AVN vs 16% (7/45) of non-AVN ($p = 0.026$). Logistic regression analysis demonstrated that excessive hip abduction showed an increased odds ratio of 4.65 [95% CI: 1.20, 18.06]. In 13 patients with excessive hip abductions, 6 patients (46%) exhibited global perfusion defects, whereas global

**Table 1. Risk factors of AVN: Clinical features.**

| Clinical Features | All (n = 58) | AVN (n = 13) | Non-AVN (n = 45) | OR [95% CI] | P-value |
|---|---|---|---|---|---|
| *Preoperative* | | | | | |
| Age at closed reduction (mo) mean ± SD | 10.95 ± 4.65 | 10.92 ± 4.50 | 10.96 ± 4.74 | 1.00 [0.87, 1.14] | 0.982 |
| Sex | | | | | |
| Female, n (%) | 53 | 11 (85%) | 42 (93%) | | |
| Male, n (%) | 5 | 2 (15%) | 3 (7%) | 2.55 [0.38, 7.16] | 0.337 |
| Breech presentation | 16 | 3 (23%) | 13 (29%) | 0.74 [0.18, 3.12] | 0.680 |
| Oligohydramnios | 10 | 1(8%) | 9 (20%) | 0.33 [0.04, 2.91] | 0.320 |
| Family history | 4 | 1 (8%) | 3 (7%) | 1.17 [0.11. 12.26] | 0.898 |
| Failure of Pavlik harness | 10 | 2 (15%) | 8 (18%) | 0.84 [0.16, 4.55] | 0.841 |
| *Operative* | | | | | |
| Adductor tenotomy | 29 | 9 (69%) | 20 (44%) | 2.81 [0.75, 10.4] | 0.124 |
| *Postoperative* | | | | | |
| Follow-up duration (mo) mean ± SD | 68.50 ± 28.09 | 66.32 ± 26.80 | 69.13 ± 28.72 | 1.00 [0.97, 1.02] | 0.749 |

Data are numbers of patients with percentages in parentheses for nominal variables and mean ± standard deviation for continuous variables.

perfusion defects were observed in 6 of 45 patients (13%) with a normal range of hip abduction angle (p = 0.026).

**Proximal femoral epiphysis: Ossific nucleus.**   The proximal femoral epiphyseal ossification center was not visible on MRI in 18 patients (31%): 54% (7/13) of AVN vs 24% (11/45) of non-AVN (p = 0.050). Logistic regression analysis demonstrated that the invisible secondary ossification center of the proximal femoral epiphysis exhibited an increased odds ratio of 3.61 [95% CI: 1.00, 13.04].

**Proximal femoral epiphysis: Contrast enhancement pattern.**   Global decreased enhancement of the proximal femoral epiphysis was observed in 12 patients (21%): 77% (10/13) of AVN vs 4% (2/45) of non-AVN (p < 0.001) (Fig 6). Logistic regression analysis demonstrated that global decreased enhancement showed an increased odds ratio of 71.66 [95% CI: 10.54, 487.31]. Interrater agreement of contrast enhancement pattern was substantial between the two readers (Kappa 0.66, 95% CI [0.52, 0.80]).

## Multi-variable analysis

Based on the clinical importance and results from the univariable analysis, the following imaging factors were included in the multi-variable model: eccentric reduction, excessive abduction, invisible secondary ossification center of the femoral head, and global decreased enhancement of the proximal femoral epiphysis. Multi-variable analysis for imaging parameters indicated that, independent of other parameters, global decreased enhancement had a significantly higher risk of developing proximal femoral growth disturbances or AVN after closed reduction for DDH (odds ratio: 27.92, 95% CI [4.17, 350.18], p = 0.0031) (Table 3). The optimism corrected area-under-the-curve was 0.84 [95% CI: 0.71, 0.97].

Results of the hip MRI analysis are presented in Tables 2 and 3.

## Discussion

According to our study results, global decreased enhancement of the proximal femoral epiphysis on MRI is a unique imaging marker to predict development of AVN after closed reduction of DDH. The cartilaginous proximal femoral epiphysis shows a speckled or striped pattern of enhancement after intravenous administration of contrast materials on MRI, which represents non-anastomosing vascular canals mainly supplied by medial femoral circumflex artery [23,24]. There were still debate regarding predictive value of contrast enhancement of the femoral head in immediate post op. contrast enhanced MRI after closed reduction. Haruno et al [17] suggested cut-off value of epiphyseal enhancement, which was less than 80%, exhibited a sensitivity of 87.5% and specificity of 88.25% in the diagnosis of epiphyseal osteonecrosis after closed reduction of DDH using contrast-enhanced spica MRI. However, Nguyen et al [25] reported that neither of enhancement pattern nor percent enhancement predicted AVN on postoperative MRI. Considering transient ischemia of the proximal femoral epiphysis induced by hip abduction, we subjectively estimated global decreased enhancement when the non-enhancing area of the proximal femoral epiphysis was larger than 3/4 of the proximal femoral epiphysis area with loss of normal striated or speckled enhancement. Among 12 patients of global decreased enhancement, 10 patients (77%) developed AVN of the proximal femoral epiphysis while 43 (96%) out of 46 patients with normal or geographic enhancement of the femoral head did not show AVN of the femoral head in this study.

The presence of an ossific nucleus at the time of reduction has been considered to be protective against osteonecrosis in some series. In our study, 7 of 18 patients with no visible secondary ossification center of the proximal femoral epiphysis developed AVN with borderline significance (P = 0.05) compared between the AVN and non-AVN groups. Some authors

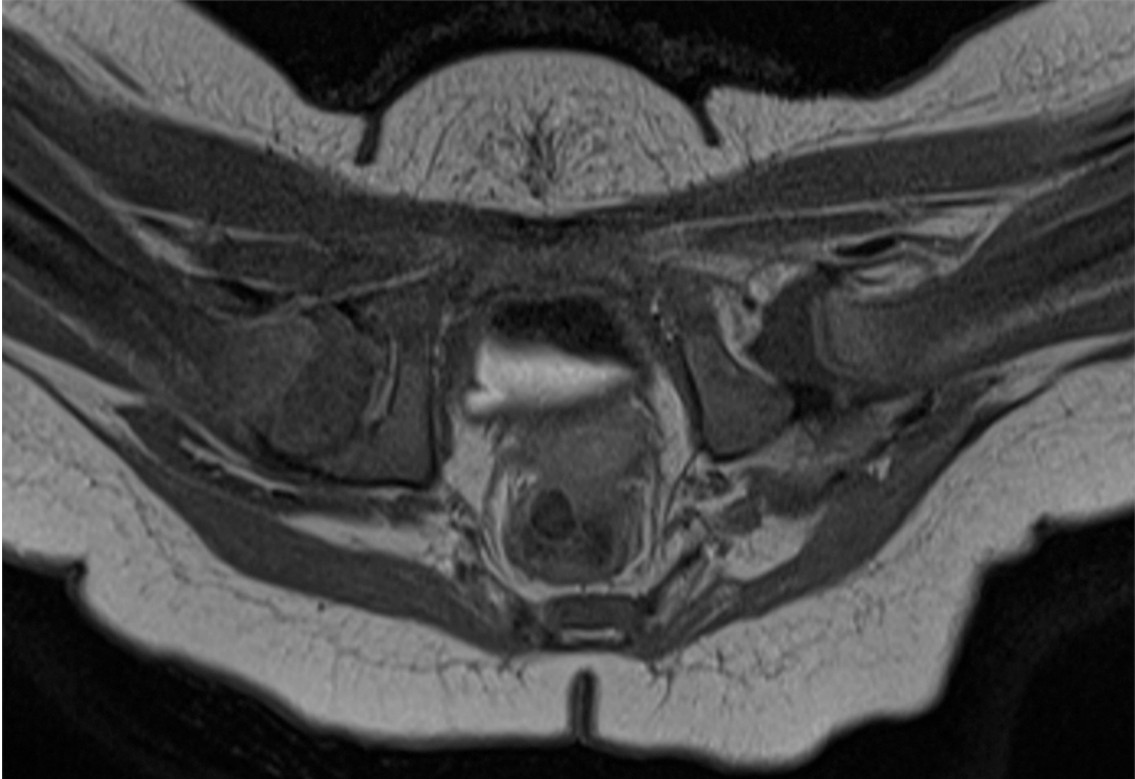

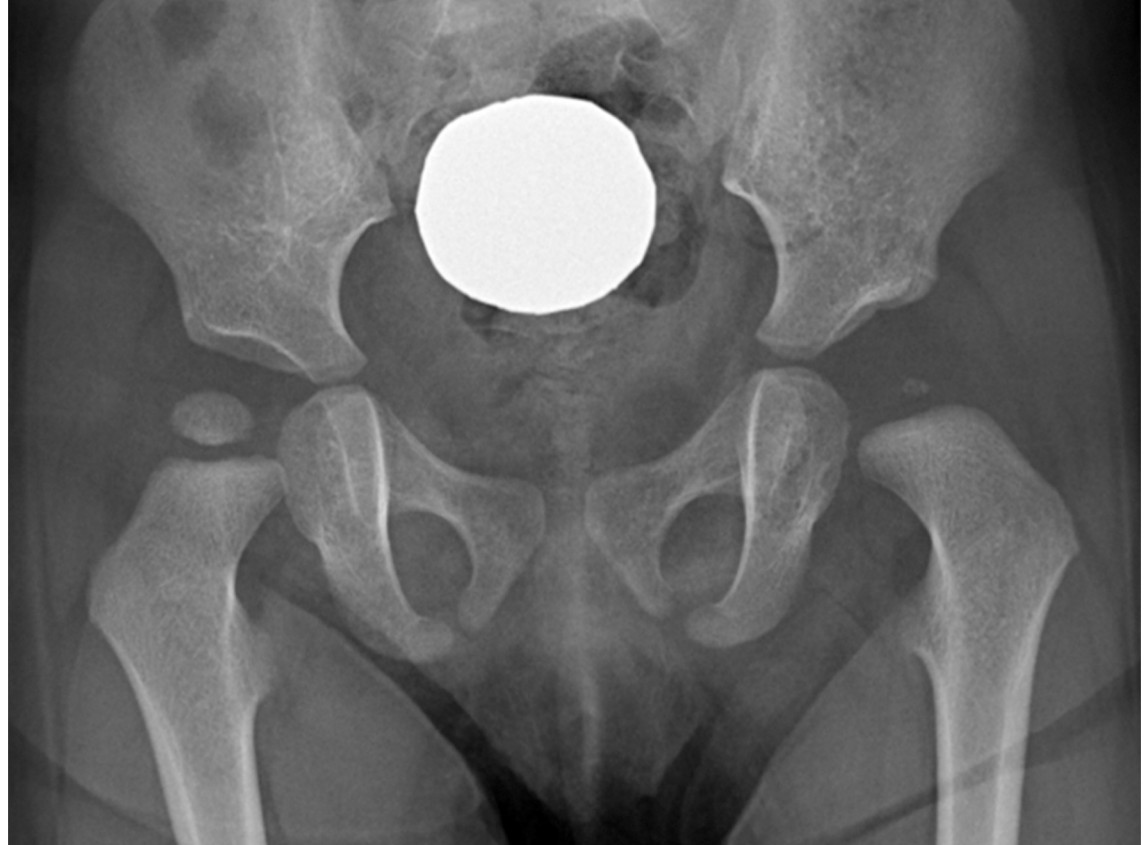

**Fig 6. Contrast enhancement of the proximal femoral epiphysis and development of AVN after closed reduction of DDH.** A. Contrast-enhanced axial T1-weighted image of a 7-month-old girl after closed reduction of the left hip shows a global decreased enhancement of the left proximal femoral epiphysis. Note the pulvinar in the left hip joint. B. Follow-up radiograph obtained 1 year after closed reduction shows a small, fragmented secondary ossification center of the proximal femoral epiphysis and widening of the femoral neck suggesting AVN.

reported an increased risk of development of AVN if reduction was performed in the absence of the ossific nucleus [5,26]. However, delaying reduction of the dislocated hip in the absence of the ossific nucleus may increase the risk for residual acetabular dysplasia avoiding the time period of maximal adaptive growth of the acetabulum [27].

Previous studies have connected excessive hip abduction with increased risk of AVN [9,17,21]. An excessive hip abduction angle has been assumed to compromise the blood supply to the femoral head by either compressing the medial circumflex femoral artery or by exerting pressure on the femoral head related to hip incongruence [8]. Recently, axial images from postoperative CT or MRI are often used to measure abduction angle in DDH after spica cast [20,21,28]. Since Smith et al. [28] reported a significant risk of subsequent development of AVN when the hip abduction angle on post-reduction CT scan was $> 55°$, it has been accepted that excessive hip abduction (angle $> 55–60°$) in spica is an important risk factor for future development of AVN. In this study, we measured hip abduction angle on axial MRI images under the assumption that hip flexion angle should be 90°. At 90° hip flexion, an axial image is orthogonal to the plane of hip flexion, therefore it is appropriate to measure abduction angle on axial plane. If the hip flexion angle is less than 90°, axial images are not orthogonal to the plane of hip flexion and it should be considered that abduction angle might be overestimated [29].

Adequacy and stability of the reduction should be evaluated based on the findings of physical examinations under anesthesia and various criteria on intraoperative arthrography [30,31]. A recent study explained that medial joint space measurements on arthrography and MRI were well-correlated, and medial joint space width less than 16% of femoral head width may be a useful intraoperative criterion for acceptable hip reduction [30]. Our study demonstrated that the AVN group had a wider medial joint space than the non-AVN group at the time of closed reduction. A recent study with follow-up MRI after initial closed reduction demonstrated that the femoral head position within the acetabulum improved over a short time

**Table 2. Risk factors of AVN: MRI features.**

| MRI Risk Factors | All (n = 58) | AVN (n = 13) | Non-AVN (n = 45) | OR [95% CI] | P- value |
|---|---|---|---|---|---|
| *Pre-contrast images* | | | | | |
| Medial joint space distance (mm) | 4.64 ± 1.32 | 5.46 ± 1.27 | 4.47 ± 1.26 | NA* | 0.014 |
| Eccentric reduction (> 3 mm of medial joint space distance) | 48 | 12 (92%) | 36 (80%) | 3.00 [0.34, 26.18] | 0.320 |
| Invisible secondary ossification center of proximal femoral epiphysis | 18 | 7(54%) | 11 (24%) | 3.61 [1.00, 13.04] | 0.050 |
| Pulvinar (+) | 50 | 13(100%) | 37(82%) | 6.12 [0.28, 134.27] | 0.251 |
| Neolimbus (+) | 37 | 9 (69%) | 28(62%) | 1.37 [0.36, 5.13] | 0.644 |
| Inversion of the acetabular labrum (+) | 2 | 0 | 2 | 0.64 [0.02, 27.94] | 0.819 |
| Excessive abduction (> 60°) | 13 | 6 (46%) | 7(16%) | 4.65 [1.20, 18.06] | 0.026 |
| *Post-contrast images* | | | | | |
| Proximal femoral epiphysis | | | | | <0.001 |
| Global decreased enhancement | 12 | 10 (77%) | 2(4%) | 71.66 [10.54, 487.31] | |
| Normal or geographic enhancement | 46 | 3 (23%) | 43 (96%) | | |

*NA: Not available.

**Table 3. Multiple regression analysis of risk factors of AVN.**

| MRI Risk Factors | OR [95% CI] | P-value |
|---|---|---|
| Eccentric reduction (> 3 mm of medial joint space distance) | 1.81 [0.18, 55.41] | 0.6505 |
| Invisible secondary ossification center of the femoral head | 1.08 [0.09, 9.00] | 0.9462 |
| Excessive abduction | 1.97 [0.21, 17.10] | 0.5283 |
| Global decreased enhancement of the proximal femoral epiphysis | 27.92 [4.17, 350.18] | 0.0031 |

period (< 1 month) following initial closed reduction [32]. This indicates that the femoral head can "dock" deeper into the acetabulum after closed reduction for DDH. Under such conditions, excessive pressure would be enforced on the femoral head during this docking period, possibly increasing the risk of AVN. Nevertheless, so-called eccentric reduction defined as more than 3 mm of medial joint space distance was not a useful imaging marker in our patients. In fact, the majority of our cases showed eccentric reduction at the time of closed reduction regardless of later development of AVN.

On the basis of our study results, global decreased enhancement of the femoral head on contrast-enhanced hip MRI is a unique imaging marker that suggests that the involved femoral head is at higher risk of developing AVN. Theoretically, cast revision with an acceptable abduction angle while maintaining closed reduction or conversion to open reduction may improve femoral head perfusion. However, early intervention or cast revision in patients with compromised proximal femoral epiphyseal perfusion after closed reduction of DDH was not a routine protocol at our institution. All of our patients underwent scheduled cast revision within 6 weeks. Perfusion of the epiphysis is a more dynamic process, and development of AVN could be related to degree of perfusion reduction with possible dependence on duration. In this point of view, if femoral head perfusion can be non-invasively monitored during spica casting, our practice routine can be altered for select cases that require early intervention to prevent AVN after closed reduction of DDH. Recently, intraoperative contrast-enhanced ultrasound (CEUS) has been suggested as a useful modality for evaluation of femoral head perfusion in patients with closed or open reduction of DDH [33]. Furthermore, a prospective study is necessary to determine whether early intervention or cast revision prevents development of AVN in patients with global decreased enhancement on hip MRI after closed reduction of DDH.

There were several limitations in this study. First, this study was retrospective study from a single institution with a relatively small number of study subjects with only a small number of patients who developed AVN. Each institution may have different indications and protocols for closed reduction and casting treatment for DDH. An additional multicenter study is necessary to expand the study population and obtain data that can aid in creating evidence-based recommendations for reducing the risk of proximal femoral growth disturbance or AVN. Second, all examinations were performed with sedation. It is necessary to develop fast MRI protocols to reduce sedation times or eliminate them all together for MRI examination and to establish a fast track to perform MRI examinations immediately after closed reduction. Third, AVN at completion of skeletal maturation should be evaluated. AVN of the femoral head was determined from radiographs obtained at 1-year post-reduction by Salter criteria in this study. Although the Salter criteria determined one to two years after reduction were reportedly predictive of the Kalamchi classification of AVN in patients with DDH aged 10 years or older [34], proximal femoral growth disturbance or AVN should be evaluated at completion of skeletal maturity.

## Conclusion

Contrast-enhanced hip MRI provides accurate anatomical assessment of the hip after closed reduction of DDH. Global decreased enhancement of the femoral head could be used as a good predictor for future development of AVN after closed reduction of DDH.

## Acknowledgments

The authors would like to thank professor emeritus In-One Kim and In Ho Choi for their support to develop concept of this study.

The authors would like to thank MRCC for statistical support.

## Author Contributions

**Conceptualization:** Jung-Eun Cheon, Ji Young Kim, Young Hun Choi, Woo Sun Kim, Tae-Joon Cho, Won Joon Yoo.

**Data curation:** Jung-Eun Cheon, Ji Young Kim, Young Hun Choi, Woo Sun Kim, Won Joon Yoo.

**Formal analysis:** Jung-Eun Cheon, Young Hun Choi, Woo Sun Kim, Won Joon Yoo.

**Investigation:** Jung-Eun Cheon, Ji Young Kim, Young Hun Choi, Won Joon Yoo.

**Methodology:** Jung-Eun Cheon, Young Hun Choi, Tae-Joon Cho, Won Joon Yoo.

**Supervision:** Woo Sun Kim, Tae-Joon Cho.

**Validation:** Jung-Eun Cheon, Young Hun Choi, Won Joon Yoo.

**Writing – original draft:** Jung-Eun Cheon, Won Joon Yoo.

**Writing – review & editing:** Jung-Eun Cheon, Ji Young Kim, Young Hun Choi, Woo Sun Kim, Tae-Joon Cho, Won Joon Yoo.

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
