## [Decision Letter · Decision Letter 0]

2 Nov 2020

PONE-D-20-28474

MRI Features and Risk of Avascular Necrosis after Closed Reduction of Developmental Dysplasia of the Hip: Predictive Value of Contrast-Enhanced MRI

PLOS ONE

Dear Dr. Yoo,

Thank you for submitting your manuscript to PLOS ONE. After careful consideration, we feel that it has merit but does not fully meet PLOS ONE’s publication criteria as it currently stands. Therefore, we invite you to submit a revised version of the manuscript that addresses the points raised during the review process.

We look forward to receiving your revised manuscript.

Kind regards,

James G. Wright

Academic Editor

PLOS ONE

Journal Requirements:

2. Please ensure that you include a title page within your main document. ** You should list all authors and all affiliations as per our author instructions and clearly indicate the corresponding author **.

3.  Thank you for submitting the above manuscript to PLOS ONE. During our internal evaluation of the manuscript, we found significant text overlap between your submission and the following previously published works:

- https://online.boneandjoint.org.uk/doi/full/10.1007/s11832-016-0743-7

- https://pubs.rsna.org/doi/full/10.1148/rg.2016150159

Please revise the manuscript to rephrase the duplicated text, cite your sources, and provide details as to how the current manuscript advances on previous work. Please note that further consideration is dependent on the submission of a manuscript that addresses these concerns about the overlap in text with published work.

Reviewers' comments:

Reviewer's Responses to Questions

**Comments to the Author**

1. Is the manuscript technically sound, and do the data support the conclusions?

Reviewer #1: Yes

Reviewer #2: Partly

2. Has the statistical analysis been performed appropriately and rigorously? 

Reviewer #1: Yes

Reviewer #2: No

3. Have the authors made all data underlying the findings in their manuscript fully available?

Reviewer #1: Yes

Reviewer #2: Yes

4. Is the manuscript presented in an intelligible fashion and written in standard English?

Reviewer #1: Yes

Reviewer #2: Yes

5. Review Comments to the Author

Reviewer #1: The subject of this study is interesting and relevant to paediatric orthopaedic surgeons. There is currently no "gold-standard" imaging that can safely predict the development of AVN of the femoral epiphysis following closed reduction for DDH. This study suggests an important first step in developing a gold standard. A few suggestions on improving the manuscript further follow.

Firstly, it would be interesting to add some information on the safety of the process. Have there been any reactions or side effects related to the administration of the contrast? Second, please can you elaborate further on why infants had to be sedated for the MRI? Does the hip spica cast not provide sufficient immobility for the MRI to be undertaken.

The Authors have done a great job in designing the study as best as possible, considering this was a single-centre trial with relatively small numbers and limited follow-up. The weaknesses of the study are recognised and adequately discussed in the Discussion section. Some further discussion of the following points would be helpful for the reader:

1. The confounding factors (previous treatment with harness/brace, traction and adductor tenotomy) were discussed and their predictive value calculated. However, the numbers were perhaps too small to provide enough power for these calculations. Could the Authors comment on this issue? Would it not be more appropriate to just focus on the predictive value of MRI?

2. How well were the 3 patterns of contrast enhancement distinguishable? How easy was to define the 80% cut-off point?

3. Is the abduction angle not affected by the position of the infant in the scanner? For example a more flexed or extended position of the pelvis resulting from the position of the infant in the spica and in the scanner would surely affect the abduction angle?

4. I am impressed by the consistency achieved in the usage of the Salter criteria for AVN. These criteria would probably cover the more severe AVN's. However, the lateral AVN causing tilting of the femoral epiphysis and coxa valga may not be evident for several years. It is claimed that the Salter criteria can predict Kalamchi classification at age 10 or older. Could the Authors expand on how solid this evidence is and comment on this issue further?

I fully agree that these results require to be confirmed in multi-centre settings with longer follow-up. This paper is a positive step in this direction.

The paper is well written and the text flows in a logical sequence. The illustrations are useful and appropriate.

Reviewer #2: This is important work and although the reference standard test is somewhat limited because it is defined as “AVN determined by Salter’s criteria at only 12 months”, the study shows that there is indeed an important value of the test. However, despite of its limitations, it is robust enough and there was supportive data on its reliability. (The key issue in interpreting the results will be to consider scenarios of varying reliability in this measure, the more unreliable it is, the less robust the results can be; however the strong OR associated with the index test suggests that reliability is good enough to denote robustness of test).

My only real concern relates to the statistical approach. I would suggest that, in order to establish criterion validity, one would only need to focus on the index test, MRI, and consider important confounders in this very analysis. (The approach presented in the paper appears that there were candidate predictors searched for and tested, which is not exactly the same.)

The key challenge is that there were only 13 events and the OR is inflated therefore because of overfitting the model. I would use penalised logistic regression instead; I would then calibrate the OR using bootstrapping and come up with an optimism adjusted area-under-the-curve.

In revising this, it would be required to state more explicitly which covariates the final model included. (And how these were chosen).

An alternative, but less useful, approach is to demonstrate correlation between index and standard reference tests.

I think that the topic is timely and it would be excellent if the authors could employ advances statistical methods to deal with the problem of small event rate and risk of model overfitting. I would also suggest to distinguish clearer the issues of (1) establishing criterion validity of a novel index test, MRI and (2) identifying predictors for “AVN at 1 year as defined by Salter”. These are 2 different issues and should not be confused.

6. PLOS authors have the option to publish the peer review history of their article (what does this mean?). If published, this will include your full peer review and any attached files.

Reviewer #1: **Yes: **Tim Theologis

Reviewer #2: No

---

## [Author Response · Author response to Decision Letter 0]

5 Jan 2021

Journal Requirements:

1. Please ensure that your manuscript meets PLOS ONE's style requirements, including those for file naming.  confirm

2. Please ensure that you include a title page within your main document. ** You should list all authors and all affiliations as per our author instructions and clearly indicate the corresponding author **. confirm

3. Thank you for submitting the above manuscript to PLOS ONE. During our internal evaluation of the manuscript, we found significant text overlap between your submission and the following previously published works: Redundant descriptions in previous manuscript were checked using 'iThenticate' program and revised it.

Response to reviewer's comments

Response to Reviewers

Reviewer #1: The subject of this study is interesting and relevant to paediatric orthopaedic surgeons. There is currently no "gold-standard" imaging that can safely predict the development of AVN of the femoral epiphysis following closed reduction for DDH. This study suggests an important first step in developing a gold standard. A few suggestions on improving the manuscript further follow.

1. Firstly, it would be interesting to add some information on the safety of the process. Have there been any reactions or side effects related to the administration of the contrast? 

Thank you for your comments. There have been no adverse reaction related to the gadolinium contrast materials in this study population.

2. Second, please can you elaborate further on why infants had to be sedated for the MRI? Does the hip spica cast not provide sufficient immobility for the MRI to be undertaken.

Regarding sedation issue, it is related to limited MRI resources in our hospital. It is very difficult to adjust scanning schedule soon after mean or immediate after closed reduction. Therefore it is our routine practice to scanning MRI after sedation in children younger than 8-year-old.

3. The Authors have done a great job in designing the study as best as possible, considering this was a single-centre trial with relatively small numbers and limited follow-up. The weaknesses of the study are recognised and adequately discussed in the Discussion section. 

Thank you for your comments.

Some further discussion of the following points would be helpful for the reader:

4. The confounding factors (previous treatment with harness/brace, traction and adductor tenotomy) were discussed and their predictive value calculated. However, the numbers were perhaps too small to provide enough power for these calculations. Could the Authors comment on this issue? Would it not be more appropriate to just focus on the predictive value of MRI?

Thank you for your comments. There have been no adverse reaction related to the gadolinium contrast materials in this study population.

Thank you for your comments. We revised the manuscript focused on MRI risk factors.

We revised manuscript title as well as results according to your comments. 

5. How well were the 3 patterns of contrast enhancement distinguishable? How easy was to define the 80% cut-off point?

We qualitatively evaluated femoral head enhancement pattern. We divided the femoral head into four quadrants. If the non-enhancing areas are greater than 3/4 (75%, it was rounded off to 80%), it was considered global decreased enhancement. We revised this description as it was.

6. Is the abduction angle not affected by the position of the infant in the scanner? For example a more flexed or extended position of the pelvis resulting from the position of the infant in the spica and in the scanner would surely affect the abduction angle?

It is a wonderful question. Hip abduction angle measure on axial images can be affected by hip flexion state. We added this issue in the discussion.

7. I am impressed by the consistency achieved in the usage of the Salter criteria for AVN. These criteria would probably cover the more severe AVN's. However, the lateral AVN causing tilting of the femoral epiphysis and coxa valga may not be evident for several years. It is claimed that the Salter criteria can predict Kalamchi classification at age 10 or older. Could the Authors expand on how solid this evidence is and comment on this issue further?

I agree you opinion. We also reviewed all postoperative radiographs in this patients. 

If there were disagreement between two readers, final interpretation was determined using a combination of Salter criteria and Kalamchi classificaxtion in consensus.

8. I fully agree that these results require to be confirmed in multi-centre settings with longer follow-up. This paper is a positive step in this direction.

The paper is well written and the text flows in a logical sequence. The illustrations are useful and appropriate.

Thank you for your comments.

Reviewer #2

This is important work and although the reference standard test is somewhat limited because it is defined as “AVN determined by Salter’s criteria at only 12 months”, the study shows that there is indeed an important value of the test. However, despite of its limitations, it is robust enough and there was supportive data on its reliability. (The key issue in interpreting the results will be to consider scenarios of varying reliability in this measure, the more unreliable it is, the less robust the results can be; however the strong OR associated with the index test suggests that reliability is good enough to denote robustness of test).

My only real concern relates to the statistical approach. I would suggest that, in order to establish criterion validity, one would only need to focus on the index test, MRI, and consider important confounders in this very analysis. (The approach presented in the paper appears that there were candidate predictors searched for and tested, which is not exactly the same.)

The key challenge is that there were only 13 events and the OR is inflated therefore because of overfitting the model. I would use penalised logistic regression instead; I would then calibrate the OR using bootstrapping and come up with an optimism adjusted area-under-the-curve.

In revising this, it would be required to state more explicitly which covariates the final model included. (And how these were chosen).

An alternative, but less useful, approach is to demonstrate correlation between index and standard reference tests.

I think that the topic is timely and it would be excellent if the authors could employ advances statistical methods to deal with the problem of small event rate and risk of model overfitting. I would also suggest to distinguish clearer the issues of (1) establishing criterion validity of a novel index test, MRI and (2) identifying predictors for “AVN at 1 year as defined by Salter”. These are 2 different issues and should not be confused.

Thank you for your valuable comments. 

As the predictors for AVN in infants who underwent closed reduction and casting for treatment of DDH are not well-established, we explored possible predictors of AVN in this study. Therefore, the objective of this study is exploring predictors of AVN, including findings of contrast-enhanced MRI. We explored candidate predictors of AVN in the uni-variable analysis and the multi-variable model was constructed based on the clinical importance and findings from uni-variable analysis.

We used penalized logistic regression to reduce sparse data bias in the analysis, however, we made mistake while writing the manuscript and corrected accordingly. Also, we presented the optimism adjusted area-under-the-curve based on 1000 bootstrap samples for the multi-variable model as you recommended.

---

## [Decision Letter · Decision Letter 1]

18 Jan 2021

PONE-D-20-28474R1

MRI risk factors for development of avascular necrosis after closed reduction of developmental dysplasia of the hip: Predictive value of contrast-enhanced MRI

PLOS ONE

Dear Dr. Yoo,

Thank you for submitting your manuscript to PLOS ONE. After careful consideration, we feel that it has merit but does not fully meet PLOS ONE’s publication criteria as it currently stands. Therefore, we invite you to submit a revised version of the manuscript that addresses the points raised during the review process.

We look forward to receiving your revised manuscript.

Kind regards,

James G. Wright

Academic Editor

PLOS ONE

Additional Editor Comments (if provided):

The revision has made presentation much clearer. One reviewer has concerns about statistical analyses and because conclusions rest on that analyses, I would ask authors to carefully review and respond and revise if needed

Reviewers' comments:

Reviewer's Responses to Questions

**Comments to the Author**

1. If the authors have adequately addressed your comments raised in a previous round of review and you feel that this manuscript is now acceptable for publication, you may indicate that here to bypass the “Comments to the Author” section, enter your conflict of interest statement in the “Confidential to Editor” section, and submit your "Accept" recommendation.

Reviewer #1: All comments have been addressed

Reviewer #2: (No Response)

2. Is the manuscript technically sound, and do the data support the conclusions?

Reviewer #1: Yes

Reviewer #2: No

3. Has the statistical analysis been performed appropriately and rigorously? 

Reviewer #1: Yes

Reviewer #2: No

4. Have the authors made all data underlying the findings in their manuscript fully available?

Reviewer #1: Yes

Reviewer #2: Yes

5. Is the manuscript presented in an intelligible fashion and written in standard English?

Reviewer #1: Yes

Reviewer #2: No

6. Review Comments to the Author

Reviewer #1: The Authors have adequately addressed my critique and I am happy that the paper is ready for publication.

Reviewer #2: Penalised logistic regression was used but there was no reference to the methods employed. Further, I was curious to learn that the SPSS statistical package can do this type of analysis; in my experience it can be done in SAS, STATA and R only (but I could be wrong). The same for the bootstrapping; also here there was no reference given about the methods used. I recommend adding references.

Statistical section should include a clear description of how the modelling was done, i.e. what were candidate predictors (it seems that there is a 2nd section in the paper entitled “Multi variable analysis” but it would better be part of the section “Statistical analysis”.

It appears that most of the risk factors shown in table 2 were nonsignificant, yet they were subjected to a multivariable analysis – and remained nonsignificant. I don’t quite follow that approach (because the aim is not to develop a prediction rule but to identify independent risk factors).

There were 13 AVN events (at hip level). There were five predictors included in the penalised logistic regression model, one of which was “eccentric reduction” but I could not find a definition of this variable.

It was a bit vague as to how these variables ended up in the final model. But what I understand was that there were several potential predictors, and 5 were selected for a multivariable model – and only 1 remained predictive of AVN. Unfortunately, this was the variable “global decreased enhancement of the proximal femoral epiphysis seen on MRI”. This, in turn, is not new knowledge or a surprise because this situation has long been known to indicate AVN.

Is your key message, therefore, is that no MRI-based predictors for AVN could be identified in this study - see table 3. It is already well-established that a lack of contract enhancement of the femoral head is a diagnostic MRI-criterion of AVN of the femoral head (at any age, also very young infants). Was it not that your study merely confirmed this finding, but did show that other MRI criteria cannot predict X-ray-AVN?

Therefore, the conclusion should reflect this. It looks to me that several MRI criteria were not helpful in predicting AVN, only one MRI criterion was.

7. PLOS authors have the option to publish the peer review history of their article (what does this mean?). If published, this will include your full peer review and any attached files.

Reviewer #1: No

Reviewer #2: No

---

## [Author Response · Author response to Decision Letter 1]

1 Mar 2021

Reviewer #1: The Authors have adequately addressed my critique and I am happy that the paper is ready for publication.

Thank you for your comments.

Reviewer #2: 

1. Penalised logistic regression was used but there was no reference to the methods employed. Further, I was curious to learn that the SPSS statistical package can do this type of analysis; in my experience it can be done in SAS, STATA and R only (but I could be wrong). The same for the bootstrapping; also here there was no reference given about the methods used. I recommend adding references.

Thank you for your comments. We address the reference [22] (Firth D. Bias reduction of maximum likelihood estimates. Biometrika 1993; 80:27–38) and revised the statistical package that we used in this study (SAS)

2. Statistical section should include a clear description of how the modelling was done, i.e. what were candidate predictors (it seems that there is a 2nd section in the paper entitled “Multi variable analysis” but it would better be part of the section “Statistical analysis”.

Potential candidate risk factors for AVN considered in this study were clinical features including demographical characteristics and MRI features. According to your previous suggestion, we focused on risk factors analysis of index test, MRI.

3. It appears that most of the risk factors shown in table 2 were nonsignificant, yet they were subjected to a multivariable analysis – and remained nonsignificant. I don’t quite follow that approach (because the aim is not to develop a prediction rule but to identify independent risk factors).

In table 2, excessive abduction (p=0.026) and global decreased enhancement of the femoral head (p <0.001) showed statistically significant difference between AVN and non-AVN groups. Invisible secondary ossification center of the femoral head showed borderline significance (p=0.05). Containment of the femoral head has been considered as a risk factor to develop AVN, therefore, ‘eccentric reduction’ which was defined as a medial joint space widening more than 3 mm was included as a candidate predictor in multi-variable analysis. 

There were four predictors in the penalised logistic regression model.

4. There were 13 AVN events (at hip level). There were five predictors included in the penalised logistic regression model, one of which was “eccentric reduction” but I could not find a definition of this variable.

It was an error to describe Table 3. There were four predictors in the penalised logistic regression model. Eccentric reduction was defined as > 3 mm of medial joint space distance. 

5. It was a bit vague as to how these variables ended up in the final model. But what I understand was that there were several potential predictors, and 5 were selected for a multivariable model – and only 1 remained predictive of AVN. Unfortunately, this was the variable “global decreased enhancement of the proximal femoral epiphysis seen on MRI”. This, in turn, is not new knowledge or a surprise because this situation has long been known to indicate AVN.

There were still debate regarding predictive value of contrast enhancement of the femoral head in immediate post op. contrast enhanced MRI after closed reduction. Recently Nguyen et al. reported that neither of enhancement pattern nor percent enhancement predicted AVN on postoperative MRI (Nguyen, Jie C., et al. "Developmental dysplasia of the hip: can contrast-enhanced MRI predict the development of avascular necrosis following surgery?." Skeletal Radiology 50.2 (2021): 389-397)

6. Is your key message, therefore, is that no MRI-based predictors for AVN could be identified in this study - see table 3. It is already well-established that a lack of contract enhancement of the femoral head is a diagnostic MRI-criterion of AVN of the femoral head (at any age, also very young infants). 

As I mentioned above, there were still debate regarding predictive value of contrast enhancement of the femoral head in immediate post op. contrast enhanced MRI after closed reduction. Geographic decreased enhancement with double rim sign is a well-known MRI features of AVN. But global decreased enhancement of the femoral head at immediate post op. MRI after closed reduction of DDH does not directly mean AVN. It is merely decreased contrast enhancement but it could be a risk factor of future AVN. Other MRI risk factors such as eccentric reduction, invisible secondary ossification center, and excessive abduction were not well-known MRI risk factors. Although both of eccentric reduction and invisible secondary ossification center were suggestive risk factors on X-ray and excessive abduction was a risk factor on clinical examination. 

7. Was it not that your study merely confirmed this finding, but did show that other MRI criteria cannot predict X-ray-AVN? Therefore, the conclusion should reflect this. It looks to me that several MRI criteria were not helpful in predicting AVN, only one MRI criterion was.

Both of eccentric reduction and invisible secondary ossification center were suggestive risk factors on radiographic examination after closed reduction of DDH. Excessive abduction was also a well-known risk factor on clinical examination after closed reduction of DDH. In this study using MRI, contrast enhancement pattern could provide information regarding increased risk of future development of AVN.

---

## [Editor Report · Decision Letter 2]

4 Mar 2021

MRI risk factors for development of avascular necrosis after closed reduction of developmental dysplasia of the hip: Predictive value of contrast-enhanced MRI

PONE-D-20-28474R2

Dear Dr. Yoo,

We’re pleased to inform you that your manuscript has been judged scientifically suitable for publication and will be formally accepted for publication once it meets all outstanding technical requirements.

Kind regards,

James G. Wright

Academic Editor

PLOS ONE
---

## [Editor Report · Acceptance letter]

9 Mar 2021

PONE-D-20-28474R2 

MRI Risk Factors for development of avascular necrosis after closed reduction of developmental dysplasia of the hip: Predictive value of contrast-enhanced MRI 

Dear Dr. Yoo:

I'm pleased to inform you that your manuscript has been deemed suitable for publication in PLOS ONE. Congratulations! Your manuscript is now with our production department. 

Kind regards, 

on behalf of

Professor James G. Wright 

Academic Editor

PLOS ONE